# Tachykinins and Kisspeptins in the Regulation of Human Male Fertility

**DOI:** 10.3390/jcm9010113

**Published:** 2019-12-31

**Authors:** Víctor Blasco, Francisco M. Pinto, Cristina González-Ravina, Esther Santamaría-López, Luz Candenas, Manuel Fernández-Sánchez

**Affiliations:** 1IVI-RMA Seville, Avda. República Argentina 58, ES-41011 Seville, Spain; victor.blasco@ivirma.com (V.B.); cristina.gonzalez@ivirma.com (C.G.-R.); manuel.fernandez@ivirma.com (M.F.-S.); 2IVI Foundation, Instituto de Investigación Sanitaria La Fe (IIS La Fe), ES-46026 Valencia, Spain; 3Instituto de Investigaciones Químicas, CSIC, Avenida Americo Vespucio 49, ES-41092 Seville, Spain; francisco.pinto@iiq.csic.es (F.M.P.); luzcandenas@iiq.csic.es (L.C.); 4Departamento de Biología Molecular e Ingeniería Bioquímica, Universidad Pablo de Olavide, ES-41013 Seville, Spain; 5Departamento de Cirugía, Universidad de Sevilla, Avda. Sánchez Pizjuan S/N, ES-41009 Seville, Spain

**Keywords:** tachykinin, kisspeptin, neurokinin, male infertility, assisted reproductive technology

## Abstract

Infertility is a global disease affecting one out of six couples of reproductive age in the world, with a male factor involved in half the cases. There is still much to know about the regulation of human male fertility and thus we decided to focus on two peptide families that seem to play a key role in this function: tachykinins and kisspeptins. With this aim, we conducted an exhaustive review in order to describe the role of tachykinins and kisspeptins in human fertility and their possible implications in infertility etiopathogenesis. Many advances have been made to elucidate the roles of these two families in infertility, and multiple animal species have been studied, including humans. All of this knowledge could lead to new advances in male infertility diagnosis and treatment, but further research is needed to clarify all the implications of tachykinins and kisspeptins in fertility.

## 1. Introduction

Infertility is a condition defined as the inability for a couple to conceive after at least one year of unprotected intercourse. This pathology has become a global health issue with a general prevalence of 15%, affecting one out of six couples of reproductive age. According to global statistics, male infertility is the cause of approximately 50% of infertility cases, either as a sole cause or in combination with a female infertility factor [1]. 

Currently, assisted reproductive technology (ART) is the most reliable tool to effectively treat infertility when a male factor is present, and thanks to numerous advancements, many strategies have been developed in order to address different male infertility etiologies [1,2]. However, many aspects remain to be unveiled and the etiology of suboptimal sperm quality is still poorly understood. Many pathological agents have been described, including genetic, physiological, and environmental factors [3,4,5,6,7]. In this sense, new advances are necessary in order to fully understand the physiology of the sperm cell and find new clinical approaches to treat male infertility.

In this review, we focused on two different groups of molecules, tachykinins and kisspeptins, deeply involved in male fertility—they are also involved in female fertility, but that is beyond the scope of this work. General aspects about these molecules are discussed, as well as their role at the hypothalamus, regulating the hypothalamic–pituitary–gonadal axis. Recent research has also revealed the possible role of tachykinins and kisspeptin exerted in spermatozoa themselves, regulating their function. We review different works describing the expression of these molecules in human spermatozoa and their possible roles in these cells.

## 2. Tachykinins

The tachykinin family is one of the most conserved peptide groups in the kingdom Animalia [8,9]. This family comprises a group of regulatory peptides including substance P (SP), neurokinin A (NKA), neurokinin B (NKB), and hemokinin-1 (HK-1) [8,10,11,12,13,14]. In humans, tachykinins are encoded by three different genes: SP and NKA are expressed through alternative mRNA (messenger ribonucleic acid) splicing from the gene tachykinin precursor 1 (*TAC1*), whereas NKB is encoded by the *TAC3* gene and HK-1 is encoded by the *TAC4* gene [12,14,15,16,17]. Tachykinin effects are mediated by three receptors named tachykinin receptor 1 (NK1R), tachykinin receptor 2 (NK2R), and tachykinin receptor 3 (NK3R), encoded respectively by the genes tachykinin receptor (*TACR*)*1*, *TACR2*, and *TACR3*. They belong to the family of G protein-coupled receptors with seven transmembrane domains. NK1R is activated preferentially by SP and HK-1, NK2R by NKA, and NK3R by NKB [8,17,18,19,20,21]. However, endogenous tachykinins are not highly selective and can act as full agonists on all three receptors [8,13] (Figure 1). 

Traditionally, tachykinins have been considered as neuropeptides, as they were mainly found in the central and peripheral nervous system [22,23]. Nowadays, numerous studies have proven that they are also expressed in non-neural cells exerting regulatory roles at many different levels. In the immune system, the expression pattern of NKB and HK-1 mRNA have been determined in human lymphocytes, monocytes, neutrophils, and eosinophils [24]. The smooth muscle cells express SP and NK1R [8,25]. In the genital tract of females, tachykinins are expressed in endometrium, oviductal epithelial cells and ovarian cells (granulosa and cumulus cells) [26,27,28,29,30]. In the genital tract of males, tachykinin and their receptors are expressed in Sertoli and Leydig cells, as well as in spermatozoa [31,32]. Moreover, mutations in the *TAC3* and *TACR3* genes were found to cause hypogonadotropic hypogonadism in humans, demonstrating that NKB and NK3R play a key role in the regulation of reproduction [33] (Figure 1).

## 3. Kisspeptins

Kisspeptins are a family of structurally related peptides encoded by the kisspeptin (*KISS1*) gene. Their effects are mediated through binding to the KISS1 receptor (KISS1R), also known as GPR54 (G protein-coupled receptor 54), which is encoded by the *KISS1R* gene. Kisspeptins are mainly expressed in the hypothalamus, regulating gonadotropin releasing hormone (GnRH) secretion and, thereby, gonadotropin release by the pituitary gland [34,35,36]. In fact, humans with mutations in the *KISS1R* gene, or mice with mutations in *KISS1* or *KISS1R* genes, are affected by hypogonadotropic hypogonadism, characterized by deficient production of gonadotropins and sex steroids, which leads to an incomplete sexual maturation [37,38,39,40,41]. Recent evidence has shown that they can also be found in mammalian reproductive tissues, including the testes [42,43], the uterus [26,29,44,45,46], the oviduct [26,47], the ovary [26,27,28,48,49,50,51,52], and the placenta [53]. 

In female reproductive tissues, kisspeptins are involved in the regulation of a wide variety of processes including follicular development, oocyte maturation, ovulation, ovarian steroidogenesis, embryo implantation, and placentation [53,54,55,56]. In the case of males, kisspeptins are suggested to play important regulatory roles in spermatogenesis, testicular steroidogenesis, and spermatozoa function, as is discussed in the following sections [43,54].

## 4. Tachykinins and Kisspeptins as Regulators of the Hypothalamic–Pituitary–Gonadal Axis

The hypothalamic–pituitary–gonadal (HPG) axis plays an essential role in maintaining the normal function of the reproductive system in adults of both sexes. The HPG axis is governed by pulsatile secretion of gonadotropin-releasing hormone (GnRH) from the hypothalamus. This hormone, in turn, regulates the secretion of gonadotropins by the pituitary gland: follicle-stimulating hormone (FSH) and luteinizing hormone (LH). In addition, the secretion of GnRH, FSH, and LH is subjected to feedback regulation by hormones produced by the gonad [53]. According to evidence, neurokinin B (NKB) and kisspeptin (KISS1) are two key agents in the regulation of this axis, together with the opioid peptide dynorphin. They are all expressed in a subset of neurons present in the arcuate nucleus (or human infundibular nucleus) of the hypothalamus—the KNDy (kisspeptin/neurokinin B/dynorphin) cells [34,36,57,58]. The co-localization of these three peptides in KNDy cells has been well documented in a high number of mammals, including sheep, goats, mice, rats, and monkeys [34,57], and has been functionally demonstrated in humans [59]. Although many aspects about these three peptides and their role in KNDy cells remain unclear, they seem to be fundamental agents in the control of GnRH pulsatile secretion by GnRH neurons, as well as its feedback regulation by gonadal hormones [57,58]. 

## 5. Expression of Tachykinins and Kisspeptins in Testicular Tissues

Various studies have been published documenting the expression of tachykinins and as well as their receptors in the testes of different animal species including humans [60,61,62]. The tachykinin SP is present in Leydig and Sertoli cells of humans, mice, hamsters, marmosets, and rats, and the mRNA of the different tachykinins and tachykinin receptors has also been detected in these cells [31,32,61,63,64]. Substance P has also been found in the seminiferous tubules of rats [65], in spermatid cells, and spermatogonia of marmosets [64]. Regarding human spermatozoa, there is also evidence of tachykinins and tachykinin receptor expression [66,67,68]. Equally, tachykinin-degrading enzymes are also expressed in these cells [66,69,70,71].

As for kisspeptins and/or KISS1R, there is documented evidence of their expression in the following tissues and cell types: round spermatid cells of mice [72]; primordial germ cells, Leydig, and interstitial cells of mice [73]; human spermatozoa [43]; Sertoli cells and interstitial compartment of rhesus monkeys [42]; and Leydig cells, Sertoli cells, and all germ cells of *Pelophylax esculentus* frogs [74].

## 6. Peripheral Roles of Tachykinins and Kisspeptins in Male Fertility Regulation

### 6.1. Spermatogenesis

Little is known about the role of tachykinins in spermatogenesis. Noritake et al. [75] found that treatment of dogs with an antagonist of the three tachykinin receptors caused an inhibition of spermatogenesis and a degeneration of testes, which appeared to be mainly mediated by NK3R. In parallel, studies in NK3R null mice demonstrated that male mice had smaller testes but apparently normal spermatogenesis [76], a fact that was also observed in NKB null male mice [77]. Further studies are needed to determine the precise role of the tachykinin system in spermatogenesis.

Scientific evidence regarding different animal species suggest that kisspeptin could be involved in the regulation of spermatogenesis. A study performed in sexually immature chum mackerel showed that subcutaneous administration of kisspeptin led to spermatogenesis acceleration [78]. In another study involving the amphibian *Pelophylax esculentus*, researchers cultured their testes—obtained in their reproductive period—and observed that kisspeptin accelerated germ cell progression [79]. As for mammals, gene expression studies performed in mice revealed that KISS1/KISS1R expression initiation in testis coincides with the formation of spermatozoa [54,72]. In addition, a cross-sectional study performed in human showed that kisspeptin was present in seminal plasma. Moreover, total measured kisspeptin was positively associated with sperm concentration, total sperm number, and total mobile sperm count [80].

In contrast, studies in rats have reported an inhibitory effect of kisspeptin in spermatogenesis. Exogenous kisspeptin treatment for 30 days in male rats led to testicular degeneration, which, according to the authors, could be due to HPG axis desensitization caused by the continuous administration of kisspeptin [81]. Curiously, a new study in rats by the same authors found testicular degeneration after kisspeptin treatment for only 12 h, suggesting that the effect was not due to HPG axis desensitization but to a hyper-stimulation of the axis [82]. In a similar work, authors reported a decrease in the amount of elongated spermatids, preleptotene spermatocytes, and daily sperm production after 12 days of treatment with different doses of kisspeptin [83]. 

To summarize, evidence leads us to think that kisspeptin is someway involved in the regulation of spermatogenesis, but its role is not yet clear. Some studies suggest that kisspeptin is a local spermatogenesis inductor, whereas others suggest a central inhibitory role of this peptide via HPG axis hyper-stimulation. Molecular mechanisms behind this regulation are yet to be elucidated. In any case, kisspeptin may not be essential for spermatogenesis, as *KISS1* and *KISS1R* mutant mice conserve low levels of spermatogenesis [84]. Moreover, male patients carrying *KISS1R* mutations respond to exogenous hormonal therapy and can recover fertility [85].

### 6.2. Spermatozoa Function

There is evidence that tachykinins NKA, NKB, SP, and HK-1 are expressed in human sperm cells, both at the mRNA and protein level [66,67]. These studies have also shown the presence of the three tachykinin receptors at the protein level [67]. Experiments performed in human sperm samples have proven that tachykinins are functionally active in spermatozoa, as they are capable of improving sperm cell progressive motility. The effect on motility is quick and dependent on tachykinin concentration. We also know that this effect is mainly mediated by activation of NK1R and NK2R, thanks to studies using tachykinin receptor-selective antagonists [67]. 

Enkephalinase or neprilysin is the major peptidase that degrades tachykinins in most human tissues [86,87,88,89,90]. Local activity of peptide-signaling molecules is tightly controlled by their enzymatic degradation. Tachykinin-degrading enzymes neprilysin (NEP) and neprilysin-2 (NEP-2) are also expressed in human sperm cells at the mRNA and protein level [66,69,70,71]. Tachykinin-degrading enzyme neprilysin-2 protein is located at the equatorial segment of human spermatozoa, suggesting a role for this enzyme in sperm fertilizing capacity [66]. One study showed that sperm from NEP2 knockout mice had normal characteristics but lower fertilization capacity, and resulting embryos had a worse development [91]. Inhibition of NEP and NEP2 by phosphoramidon leads to an increase in linear and straight motility of sperm cells (sperm progressive motility), which is essential for adequate swimming through the female genital tract [92,93,94].

Regarding kisspeptins, it has been proven that KISS1 and KISS1R are present in mature human spermatozoa [43]. They are mainly located in the equatorial segment, which has an important role in oocyte-sperm fusion, and in the neck, involved in flagellar movement. Importantly, kisspeptin colocalizes with NKB in the equatorial segment [43]. Kisspeptin is able to induce a slow and sustained increase in intracellular Ca^2+^ concentration in sperm cell, which has been associated with sperm motility, hyperactivation, and acrosome reaction [93,94,95,96]. Furthermore, kisspeptin can induce changes in spermatozoa motility, increasing the flagellar beating and amplitude of lateral head displacement, which are characteristic patterns of hyperactivated spermatozoa [97]. At the same time, straight and linear movement is decreased [43]. It is well known that hyperactivated movement patterns are necessary for sperm cells in order to leave their reservoirs in the oviductal isthmus, and reach and fuse with the oocyte in vivo [94,97,98].

Besides tachykinins and kisspeptins, other bioactive peptides—such as opioids [71] and bradykinin [99]—are also expressed in spermatozoa, and many of the enzymes involved in their metabolism (like NEP and NEP-2) are also present and functionally active [69,71,99]. Inhibition of these enzymes causes changes in spermatozoa motility [66,71,99]. It is possible that these peptides work as signaling molecules between spermatozoa and their environment, acting in an autocrine and/or paracrine manner.

### 6.3. Testicular Steroidogenesis

Androgens are steroid hormones secreted by different tissues in humans, including testes, ovaries, and adrenal glands. In testes, Leydig cells are the ones responsible for androgen production and secretion. Androgens are involved in multiple processes, but their main functions include the formation of testes and male genitalia during prenatal development, the emergence and maintenance of male secondary sex characteristics in males from puberty onwards, and the support of spermatogenesis [100]. 

Different studies suggest the involvement of tachykinins or kisspeptins in androgen production by testes. Treatment of testicular sections or of isolated Leydig cells with SP or NKA reduced testosterone basal levels, as well as the increase induced by LH or hCG (human chorionic gonadotropin) [19,61,63,101]. In Sertoli cells, treatment with NKA in the presence of testosterone caused an increase in estrogen levels, suggesting an activation of aromatase [61]. 

Experiments in rats have revealed that exogenous kisspeptin administration causes an initial increase in testosterone plasma levels, but this effect vanishes if the treatment is prolonged [102]. These results were in harmony with those of Thompson and colleagues, who proposed the HPG desensitization hypothesis after kisspeptin continuous treatment, previously mentioned in this review [81]. Experiments in Leydig cells isolated from goat testes showed that a kisspeptin antagonist (P234) significantly attenuated both basal and hCG-activated testosterone and estradiol production [103]. Lastly, ex vivo experiments on testes from *P. esculentus* frogs showed that kisspeptin altered the expression of several enzymes involved in steroidogenesis, also suggesting a role for kisspeptin in steroidogenesis regulation [74]. 

However, in a different study involving mice, authors did not observe a response of Leydig cells to kisspeptin stimulation, but it is worth signaling that they used an immortalized Leydig cell line (MA-10) to perform the experiments [72]. Conditions between Leydig cells and MA-10 cell line may differ. Furthermore, steroidogenesis regulation may differ depending on the studied species. Proposed roles of tachykinins and kisspeptins in the different reviewed studies are summarized in Figure 2.

## 7. Conclusions

Kisspeptins and tachykinins are key regulators of human fertility, acting both at central and peripheral levels of the organism. On one hand, neurokinin B and kisspeptin are key regulators of HPG axis and thus gonadal function. On the other hand, tachykinins and kisspeptins are expressed in different peripheral tissues exerting regulatory functions. Much has been published about the peripheral regulatory roles of kisspeptin in female fertility. Kisspeptin and its receptor KISS1R have important roles in regulating follicle development, oocyte maturation, ovulation, ovarian steroidogenesis, implantation, pregnancy, and placentation [36,54,55,56]. However, less is known about the peripheral reproductive roles of the tachykinin family and the roles of kisspeptins and tachykinins in male reproductive tissues. 

In summary, the published work has proven that kisspeptins, tachykinins, and their corresponding receptors are expressed in testes and spermatozoa of different animal species, including humans. Evidence suggests that they have potential regulatory roles regarding spermatogenesis, spermatozoa function and motility, and testicular steroidogenesis. 

New advances are necessary in order to clarify and deepen the roles of these peptide families and increase our knowledge about the regulation of male fertility in mammals and, more importantly, in humans. These molecules could serve as genetic biomarkers to improve the diagnosis of different infertility-related diseases in men or as new targets to develop therapies to treat male infertility. Problems such as spermatogenesis defects or altered gonadal steroidogenesis, as well as alterations found in semen analysis—that is, asthenozoospermia—could be addressed in the future with treatments aimed at specific tachykinins or kisspeptins.

## Figures and Tables

**Figure 1 jcm-09-00113-f001:**
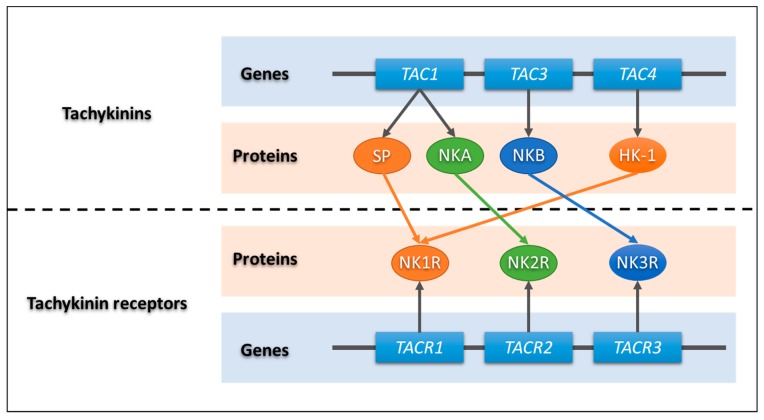
Genetic coding scheme of the tachykinins and their receptors. *TAC1*, tachykinin precursor 1 gene; *TAC3*, tachykinin precursor 3 gene; *TAC4*, tachykinin precursor 4 gene; SP, substance P; NKA, neurokinin A; NKB, neurokinin B; HK-1, hemokinin-1; *TACR1*, tachykinin receptor 1 gene; *TACR2*, tachykinin receptor 2 gene; *TACR3*, tachykinin receptor 3 gene; NK1R, tachykinin receptor 1; NK2R, tachykinin receptor 2; NK3R, tachykinin receptor 3.

**Figure 2 jcm-09-00113-f002:**
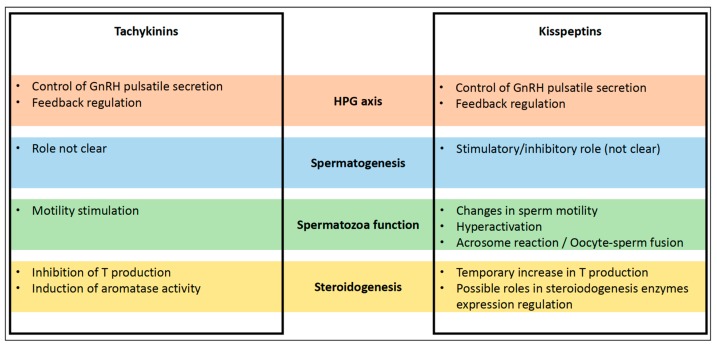
Roles of tachykinins and kisspeptins in male fertility regulation. HPG, hypothalamic–pituitary–gonadal; T, testosterone. GnRH, gonadotropin releasing hormone.

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
