# Peer review of "Tachykinins and Kisspeptins in the Regulation of Human Male Fertility"

_jcm, 2019, doi:10.3390/jcm9010113_

Round 1

Reviewer 1 Report

This is an interesting review articole foucusing on the current evidence supporting th possible role of tachykinins and kisspeptins on male fertility. Within the limits of a mini-review, the manuscript is quite complete and well written, requiring only minor English Language editing. I think that the paper would benefit of a figure summarizing the effects of tachykinins and kisspeptins on both the male reprodctiv endocrinology and on functional propertiesof sperm cells.

Author Response

Thanks for you comments. English Language has been reviewed and Figure 2 has been included following your suggestions.

Reviewer 2 Report

This is an interesting review about tachykinins and kisspeptins in the regulation of human male fertility. The article is accurately wrote, easy to understand and contains important and recent knowledge.

Comments:

L63-Please expand what exactly means immune cells, gential tract of female and male tract. Which cells?

L88- In the manuscript, there is a mess with abbreviation, this line is an example.

L102 Is this title grammatically correct?

L153 and L102 - do not start sentence with abbreviation.

Accroding to the author the knowledge about kisspeptins and tachykinins could lead to new advances in male infertility diagnosis and treatment, please expand more this issue.

Author Response

Thanks for your comments. 

L63- We have modified the previous line by "In the immune system, the expression pattern of NKB and HK-1 mRNA have been determined in human lymphocytes, monocytes, neutrophils and eosinophils [24]. The smooth muscle cells express SP and NK1R  [8,25]. In the genital tract of females, tachykinin are expressed in endometrium and oviduct by endometrial and oviductal epithelial cells, and in ovary by granulosa an cumulus cells [26-30]. In the genital tract of males, tachykinin and their receptors are expressed in Sertoli and Leydig cells and in spermatozoa [31,32]."

L88- We have included the Figure 1 to clarify these abbreviations.

L102- This title has been modified: "Substance P has also been found in the seminiferous tubules of rats [65], in spermatid cells and spermatogonia of marmosets [64]."

L153 and L102 - Both abbreviations have been replaced: "Tachykinin-degrading enzymes neprilysin-2 protein"

Accroding to the author the knowledge about kisspeptins and tachykinins could lead to new advances in male infertility diagnosis and treatment, please expand more this issue. This issue has been expanded: "New advances are necessary in order to clarify and deepen the roles of these peptide families and increase our knowledge about the regulation of male fertility in mammals –and more importantly humans. These molecules could serve as genetic biomarkers to improve the diagnosis of different infertility-related diseases in men or as new targets to develop therapies to treat male infertility. Problems like spermatogenesis defects or altered gonadal steroidogenesis, as well as alterations found in semen analysis –i.e. asthenozoospermia– could be addressed in the future with treatments aimed at specific tachykinins or kisspeptins."
